# Tungsten-Doped VO_2_/Starch Derivative Hybrid Nanothermochromic Hydrogel for Smart Window

**DOI:** 10.3390/nano9070970

**Published:** 2019-07-02

**Authors:** Yu Wang, Fang Zhao, Jie Wang, Li Li, Kaiqiang Zhang, Yulin Shi, Yanfeng Gao, Xuhong Guo

**Affiliations:** 1State Key Laboratory of Chemical Engineering, East China University of Science and Technology, Shanghai 200237, China; 2Engineering Research Center of Materials Chemical Engineering of Xinjiang Bingtuan, Key Laboratory of Materials Chemical Engineering of Xinjiang Uygur Autonomous Region, Shihezi University, Shihezi 832000, China; 3School of Materials Science and Engineering, Shanghai University, Shanghai 200444, China; 4School of Materials Science and Energy Engineering, Foshan University, Foshan 528000, China

**Keywords:** W-doped VO_2_ (M) NPs, starch derivative, inorganic and organic hybrid film, thermochromism, smart window

## Abstract

Highly efficient energy-saving windows with high solar modulation properties (Δ*T*_sol_) are the everlasting pursuit of research for industrial applications in the smart window field. Hybridization is an effective means of improving both Δ*T*_sol_ and luminous transmittance (*T*_lum_). In this paper, hybrid thermochromic films were synthesized using tungsten-doped VO_2_ nanoparticles (NPs) and starch derivatives. Thermoresponsive 2-hydroxy-3-butoxypropyl starch (HBPS) was prepared with a low critical solution temperature (LCST) varying from 32 to 21 °C by the substitution of reactive groups. The hybrid film was obtained by dispersing W-doped VO_2_ NPs in HBPS hydrogels, which exhibiting remarkable solar modulation property (Δ*T*_sol_ = 34.3%) with a high average luminous transmittance (*T*_lum, average_ = 53.9%).

## 1. Introduction

Facilities in buildings including heating, ventilation, and air conditioning use most of the world’s energy [1]. Around the world, 40% of building’s energy consumption is attributed to air conditioning [2], which induces a series of problems. Reducing the energy consumption of these facilities while maintaining the comfort level is one of the biggest challenges in the sustainable development of buildings [3]. One effective way to improve a building’s energy efficiency is coating the glass with chromogenic materials such as electrochromic, gasochromic, and thermochromic materials to obtain “smart windows” [4,5,6,7,8,9,10]. Among these materials, thermochromic smart windows show advantages over others, mainly because they can regulate the solar transmittance in a temperature-responsive manner without extra energy input.

Vanadium dioxide (VO_2_) is a commonly used inorganic thermochromic material in smart window systems. It undergoes a reversible insulator–metal transition at a critical temperature of 68 °C with a huge optical transmittance change in the infrared range (IR) [3]. Chen et al. [11] synthesized the high crystalline-quality VO_2_ nanoparticles (NPs) through a “heating-up” process, which showed a high solar modulating ability (Δ*T*_sol_ = 22.3%). In the meantime, organic thermosensitive polymers are also favorable for the smart window system, such as poly (*N*-isopropylacrylamide), i.e., PNIPAm, which is a hydrogel whose phase transition from a swollen state to aggregated state results in a prominent visual difference at its lower critical solution temperature (LCST, the temperature at 50% of the initial transmittance) [12]. By tuning the thickness of the PNIPAm hydrogel layer, Zhou et al. [13] obtained a smart window system based on pure PNIPAm film showing excellent solar modulating ability (Δ*T*_sol_ = 25.5%) with a high luminous transmittance (*T*_lum_ = 70.7%). For the sealing problem of PNIPAm hydrogel-based smart windows, a co-solvent method was introduced in our previous work [14]. Although VO_2_ and thermoresponsive polymers are the most investigated materials in smart window systems, requirements for their commercialization have not yet been totally achieved. The purpose of efficient smart windows is to maximize energy saving by increasing the *T*_lum_ and Δ*T*_sol_ values, but smart windows made from pure VO_2_ or pure PNIPAm hydrogel could not meet the actual demands.

In 2015, Long et al. integrated inorganic VO_2_ with PNIPAm for the first time, and thus opened up a new research direction for smart window systems [15]. To the best of our knowledge, this hybrid thermochromic film achieved the highest Δ*T*_sol_ (35%) in the literatures reported so far. Similarly, the ligand exchange thermochromic system was coupled with VO_2_ NPs, resulting in a hybrid film that gave almost unchanged *T*_lum_ values after the phase transition (*T*_lum, low_ = 73.36% and *T*_lum, high_ = 68.71%) with relatively high Δ*T*_sol_ values [16]. Although the hybridizing process offers both a larger visible and IR modulating ability, the high critical temperature of VO_2_ in the polymer matrix still restricts its application in smart windows. Doping by tungsten can significantly reduce the critical temperature of VO_2_ and solve the above problem [17]. Much effort has been made to investigate the effect of W doping on the phase transition temperature of VO_2_. By the thermolysis method, a critical temperature decrease of 20 K per at % W in VO_2_ nanopowders was achieved by Peng et al. [18]. Chen et al. reported a critical temperature reduction of 8 K per at % W using a “heating-up” method [11], and Zhao et al. synthesized V_0.99_W_0.01_O_2_ with a decreasing gradient of 20 K per at % W [19].

Previously, a smart window film based on thermoresponsive and pH-responsive starch derivatives has been reported [20]. In this work, tungsten-doped VO_2_/starch derivatives hybrid thermochromic films were obtained through a facile way. Firstly, temperature-responsive 2-hydroxy-3-butoxypropyl starch (HBPS) was successfully obtained through an etherification reaction. The LCST of these starch derivatives can be easily tuned by varying the degree of substitution of reactive groups to meet the different practical demands. Secondly, a near-room temperature hybrid thermochromic composite (LCST of HBPS was ~32 °C and the critical temperature of W-doped VO_2_ was ~40 °C) was synthesized, which can maximize energy saving. Thirdly, the solar modulating ability (Δ*T*_sol_), especially the IR modulating ability (Δ*T*_IR_), was improved by hybridizing HBPS with W-doped VO_2_ NPs.

## 2. Materials and Methods

### 2.1. Materials

Soluble starch, sodium tungstate, and NaOH were purchased from Sinopharm Chemical Reagent Co, Ltd. (Shanghai, China). Butyl glycidyl ether (BGE, 96%) and 3-(trimethoxysilyl) propyl methacrylate (KH570, 98%) were obtained from the Tokyo Chemical Industry Co., Ltd. (Tokyo, Japan). Polyvinyl pyrrolidone (PVP, K30, Mw = 58,000 g/mol) purchased from Aladdin Reagent (Shanghai, China) Co., Ltd. V_2_O_5_ (99.6%) was purchased from Sigma-Aldrich. H_2_C_2_O_4_·2H_2_O (AR, 99%) was purchased from Chron Chemical Reagent Co, Ltd. (Sichuan, China). Polyurethane (PU, DISPERCOLL U54) was purchased form Bayer AG (Leverkusen, Germany). All the reagents were used without further purification.

### 2.2. Preparation of HBPS and HBPS Hydrogel Films

Temperature-responsive starch HBPS was synthesized through an etherifying reaction between soluble starch and BGE in aqueous solution. The general synthetic route for the preparation of HBPS is presented as follows. Soluble starch of 4.05 g was suspended in the aqueous solution in a 100-mL three-necked flask; then, 0.5 g of NaOH particles were added to the above mixture, and the resultant mixture was heated to 75 °C in the water bath. One hour later, a certain amount of BGE was added to the flask. The reaction was carried out at 75 °C for 5 h followed by neutralizing the reaction mixture to pH 7.0 [21]. The degree of substitution (DS) by BGE groups in soluble starch was calculated from ^1^H NMR data (Table 1) using the following equation:(1)$$DS=\frac{I_{{\mathrm{CH}}_{3}}/3\;}{I_{H1}}$$
where $I_{{\mathrm{CH}}_{3}}$ is the integral for the methyl group peak of the BGE group at 0.8 ppm (Peak c in Figure 1a), and $I_{H1}$ is the integral for the anomeric proton of the starch backbone between 5.1–5.4 ppm (Figure 1a).

The LCST value of HBPS hydrogel could be controlled by varying the dosage of BGE [22]. HBPS-1 and HBPS-2 (Table 1) with different LCST values were obtained. HBPS particles were mixed in deionized water to form HBPS hydrogel in ultrasonics for 10 min to ensure thorough dispersion. Then, HBPS hydrogel was dispersed in the glass cuvette with 3-mm thickness (Figure 2) to obtain the HBPS hydrogel films.

### 2.3. Preparation of W-Doped VO_2_ NPs and VO_2_/PU Composite Film

In a typical procedure, 0.125 g of V_2_O_5_ powders were dispersed into 40 mL of 0.15 M of H_2_C_2_O_4_·2H_2_O to form a uniform dispersion. Then, sodium tungstate was added directly to the above dispersion as the doping agent, following by stirring for around 30 min. The pH value of the resultant dispersion was adjusted to 7 by using NaOH solution (0.1 M). A brown precursor was formed during the addition of NaOH. The precursor was collected, washed with deionized water for several times, and then dried in a vacuum oven. Then, the precursor powders were placed in a nitrogen-filled furnace to undergo the solid-state reaction at 60 °C for 10 h. The final nanoparticle product was collected, washed for three times with deionized water, and dried in a vacuum oven at 60 °C for 12 h. As reported by our previous paper [23], the critical temperature and the doping ratio exhibit a linear relationship. Thus, the actual W content in the as-prepared NPs with a phase transition temperature of 39 °C (Appendix A) could be determined to be approximately 1.0 at %.

Then, the as-prepared W-doped VO_2_ NPs (~2 wt %) were dispersed in the deionized water with continuous stirring for around 10 min, and a certain amount of the KH 570 was added followed by ultrasonic treatment for approximately 30 min. Then, PU was gradually added under stirring conditions for 20 min. Finally, the VO_2_/PU composite film with a 2 to 3-μm thickness was formed by casting the above suspension on the PET (polyethylene terephthalate) substrate using an automatic coating machine (Elcometer 4340, Jerrymeter Co., Ltd., Beijing, China) equipped with grooved wire rods (groove depth 2 μm) and dried at 80 °C for 1 min.

### 2.4. Preparation of W-doped VO_2_/HBPS Hydrogel Composite Films

Figure 3 is a schematic illustration of the route to prepare W-doped VO_2_/HBPS hydrogel composite films. In order to prevent W–VO_2_ NPs precipitation from liquid-like HBPS hydrogel, a known amount of W–VO_2_ NPs were firstly added into PVP solution (2 g/L) to obtain uniform dispersion, which is a common method for the modification of VO_2_ NPs [24]. After 24 h of magnetic stirring, the W–VO_2_ dispersion was mixed with HBPS hydrogel. Since there was no post-processing step, the final concentration of the polymer solution was 5 and 10 g/L, respectively. Composites 1, 2, and 3 were synthesized with different addition amounts of W–VO_2_ (see Appendix A). The composite film was obtained through dropping into the “sandwich” structure (Figure 3b) to act as the function layer of the smart window. The actual contents of HBPS and W–VO_2_ in these two composite films are shown in Appendix A.

## 3. Characterization Methods

The transmittance spectra of HBPS films and composite films under normal incidence irradiation from 350 to 2600 nm were monitored in a sealed glass cuvette with a 3-mm thickness on the UV-vis near-IR spectrometer (U-4100, Hitachi, Ltd., Tokyo, Japan), as shown in Figure 2. The transmission data was collected using an integrating sphere. In order to assess the energy-saving performance of all the samples, the integrated luminous transmittance (*T*_lum_, 380–780 nm), IR transmittance (*T*_IR_, 780–2500 nm), and solar transmittance (*T*_sol_, 240–2600 nm) were calculated with Equation (2):(2)$$T_{\mathrm{lum}/\mathrm{IR}/\mathrm{sol},\;\mathrm{low}}=\frac{\int {\varphi_{\mathrm{lum}/\mathrm{IR}/\mathrm{sol},\;\mathrm{low}}}^{(\lambda )T(\lambda )d\lambda }}{\int {\varphi_{\mathrm{lum}/\mathrm{IR}/\mathrm{sol},\;\mathrm{low}}}^{(\lambda )\lambda d}}$$

*T*(λ) denotes spectral transmittance, and $\varphi_{\mathrm{lum}}(\lambda )$ is the spectra sensitivity of the light-adapted eye. $\varphi_{\mathrm{IR}}\left(\lambda \right)$ and $\varphi_{\mathrm{sol}}(\lambda )$ are respectively the IR and solar irradiance spectrum for air mass 1.5 corresponding to the sun standing 37° above the horizon [24].

Δ*T*_lum/IR/sol_ was calculated by:
Δ*T*_lum/IR/sol_ = *T*_lum/IR/sol, low_ − *T*_lum/IR/sol, high_(3)
where subscripts ‘low’ and ‘high’ denote low and high temperature, respectively.

*T*_lum, average_ was calculated by:*T*_lum, average_ = (*T*_lum, low_ + *T*_lum, high_)/2
(4)

The response behavior of the prepared samples was monitored also by the UV-vis near-IR spectrometer with a water bath to control the system temperature. The transmittance at 1100 nm was recorded as a function of time during the switching process. Both the cooling and heating tests were conducted under the same condition. In the heating process, a sample at 15 °C was quickly moved to a water bath at 60 °C, and in the cooling process, a sample at 60 °C was rapidly transferred to a water bath at 15 °C. The heating and cooling speeds were 0.16 °C/s and 0.13 °C/s, respectively.

^1^H NMR spectra were recorded on a nuclear magnetic resonance spectrometer (Varian INOVA 400, Palo Alto, CA, USA) at room temperature. For ^1^H NMR characterization, HBPS were first dissolved in DMSO-d6 containing a few drops of D_2_O. Then, these HBPS samples were measured on an ^1^H NMR spectrometer. Fourier transform infrared (FT-IR) spectra of HBPS samples were monitored on a FT-IR spectrometer (IR-430, JASCO, Inc., Tokyo, Japan). The LCST of HBPS was measured with a UV-vis spectrophotometer (Perkin Elmer Lambda 35, PerkinElmer, Inc., Waltham, MA, USA). The content of vanadium in the composites was determined by inductively coupled plasma (ICP Inc., Thermoelectric, IRIS Intrepid, West Chester, PA, USA). Then, the actual contents of W-doped VO_2_ NPs in the composites were calculated from the ICP results (see Appendix A).

## 4. Results and Discussion

### 4.1. Structure Analysis and Thermoresponse Properties of HBPS

As we know, the LCST value, which is defined as the temperature giving 50% of the initial transmittance, can be tailored by changing the degree of substitution of reactive groups [25]. In order to meet the different demands of smart window film applications, HBPS samples with different thermoresponse properties were obtained (Table 1). The ^1^H NMR spectra clearly indicate the successful preparation of HBPS by the presence of newly emerging peaks (Figure 1a). In comparison with the unmodified starch, the new peak at 0.8 ppm (peak c) is assigned to the methyl protons of BGE. Peaks at 1.2 and 1.4 ppm and peaks between 5.1–5.4 ppm are due to the methylene group of BGE and the protons of anhydroglucose units (AGUs) of the starch, respectively [26].

Two different HBPS samples with different DS_BGE_ values (both with a HBPS content of 5 g/L), i.e., HBPS-1 and HBPS-2, were prepared to test the thermoresponsive properties of HBPS. Figure 1b shows the effect of DS_BGE_ on the phase transition behaviors of HBPS at different temperatures. The transmittance curves of the samples HBPS-1 and HBPS-2 in Figure 1b indicate that the LCST value decreased (32 to 21 °C) with increasing DS_BGE_ (0.31 to 0.42). This result clearly suggests that the hydrophobic groups can affect the thermoresponsivity of starch derivatives. When there are more hydrophobic BGE groups being grafted on the starch backbone of HBPS, the LSCT value of HBPS will be lower [26]. Since the LCST value of starch derivatives can be easily moderated over a wide range of temperature at daytime, this property can meet the various requirements of smart window systems.

### 4.2. Optical Transmittance Properties of HBPS Films

Figure 4 shows the solar light transmittance curves of HBPS-1 (DS_BGE_ = 0.31) film and HBPS-2 (DS_BGE_ = 0.42) film with various concentrations at 15 and 45 °C. It can be seen that below the LCST (at 15 °C), the aqueous solution of polymer appeared highly transparent and in a hydrophilic swollen state. When the temperature increased and phase transition occurred, the hydrogen bonds collapsed gradually between the polymer and water molecules, making the solution opaque. The increase of polymer concentration will lead to the formation of larger aggregates [27], resulting in the decrease in the transmittance of the HBPS-1 and HBPS-2 films in both the visible and IR ranges.

As illustrated in Figure 4a and Table 2, the *T*_lum, high_ of HBPS-1 film slightly decreased with increased HBPS concentration after phase transition, but still remained at a high level of 49.6% at a high concentration of 10 g/L. For the transmittance curves of HBPS-2 film (Figure 4b), there existed a large contrast in both the visible and IR rangess (e.g., *T*_lum, high_ of HBPS-2 with 5 g/L was 28.9%). The same trend can be seen in the plots of Δ*T*_lum_ (%) and Δ*T*_sol_ (%) against concentration for HBPS-1 film and HBPS-2 film (Figure 4c). At each concentration, both the values of Δ*T*_lum_ (%) and Δ*T*_sol_ (%) for the HBPS-2 film were higher than those of the HBPS-1 film. However, this high Δ*T*_lum_ (%) of the HBPS-2 film will lead to a low luminous transmission after phase transition (e.g., *T*_lum, high_ of HBPS-2 film with 10 g/L was only 7.8%). Compared with the HBPS-2 film, the HBPS-1 film was more suitable for smart windows with higher luminous transmission, and the LCST of the HBPS-1 film was 32 °C, which is also close to the phase transition temperature of W-doped VO_2_ (M) (39 °C, see Appendix A). Herein, HBPS-1 films with concentrations of 5 g/L and 10 g/L were opted for further experiments.

The optical properties of HBPS-1 film with different contents (5 and 10 g/L) were briefly investigated. As presented in Figure 5a,b and Table 2, the increase of HBPS content has a slight effect on both *T*_lum_ and Δ*T*_sol_ values. The *T*_lum_ value of the HBPS-1 film reduced from 97.9% to 95.8% with increased HBPS-1 contents. The phase transition speed test was also conducted. Traditionally, a high-phase transition speed is necessary for smart window applications. On the other hand, controlling the transmittance in a stepwise manner on demand has a lot of advantages [27], such as for example, reducing the unnecessary energy consumption and a suitable visible transmittance for views. As shown in Figure 5c, the phase transition time period of HBPS-1 (10 g/L) was around 300 s, which is higher than that of the PNIPAm microgel to some extent. This long phase time period of HBPS-1 film will cause the transmittance to change gradually (Figure 5d), thus making it favorable to realize the stepwise solar control on practical demand.

### 4.3. Optical Transmittance Properties of Composite Films

The transmittance spectra and values of *T*_lum, low_, Δ*T*_lum_, ΔT_IR_, and Δ*T*_sol_ of different composites (composites 1, 2, and 3) with different contents of W-doped VO_2_ NPs were investigated. As shown in Figure 6 and Table 2, the luminous transmittance at 25 °C and the value of *T*_lum, average_ were both reduced with increased W-doped VO_2_ NPs content. Due to the increased Δ*T*_IR_ and Δ*T*_lum_ values, the value of the solar modulation was augmented from 24.7% of Composite 1 to 32.5% of Composite 2 and 34.3% of Composite 3. For further investigation of the optical performance of the composites, composites 2 and 3 were chosen and compared to HBPS and the W-doped VO_2_/PU film.

The transmittance spectra of the pure HBPS-1 film (5 g/L), W-doped VO_2_/PU film, Composite 2 film, and Composite 3 film are shown in Figure 7a. The Δ*T*_lum_ value of the pure HBPS-1 (5 g/L) film was high (44.8%), thus leading to high solar modulation properties (28.9%). However, the IR modulation property of the pure HBPS-1 (5 g/L) film was low. As we know, VO_2_ has large transmission contrast in the IR region, so dispersing VO_2_ NPs into the polymer matrix will enhance the optical performance. Due to the large contrast in the IR range of around 1250 nm, the existence of W-doped VO_2_ improves the solar modulation property of composite films. As shown in Figure 5b and Table 2, the Δ*T*_IR_ (%) value increased from 5.0 to 8.5 with an increase in W-doped VO_2_ concentration, and the solar modulation property was also improved from 28.9% to 34.3%. Although increasing the concentration of W-doped VO_2_ lowered the transmittance at both 20 and 45 °C, the obtained composite films were superior in the modulation ability of the visible range and the infrared range. The pleasing performance of Composite 2 is favorable for smart window systems. Comparing with PNIPAm hydrogel smart window systems, the Composite 2 film has a much higher Δ*T*_sol_ value (32.5% versus 21.4%). The VO_2_/PNIPAm hydrogel was also compared with the Composite 2 film in Table 2, confirming that the Composite 2 film has a slightly higher *T*_lum, average_ (%) value with a slightly lower Δ*T*_sol_ (%) value (Table 2).

### 4.4. Cyclic Stability of Composite Films

Since the high W-doped VO_2_ NP loading capacity could cause sedimentation problems, Composite 3 was chosen for the durability test. As shown in Figure 8, only slight fluctuations in the values of Δ*T*_sol_ and *T*_lum_ were observed after eight cycles of measurement. Considering the satisfying performance stability, the suitable transition temperature, and the best optical performance exhibited by the composites prepared in this work, they should be ideal candidates for the thermochromic smart window system.

## 5. Conclusions

In this work, for the first time, a W-doped VO_2_/starch derivative thermochromic hybrid film was obtained. Firstly, we prepared pure W-doped VO_2_ (M) NPs and pure HBPS hydrogel both with a low phase transition temperature (39 and 32 °C respectively). Due to these two similar and low phase transition temperatures, the near-room temperature composite thermochromic film was obtained by incorporating W-doped VO_2_ (M) with HBPS hydrogel. It was found that these composite films with W-doped VO_2_ (M) NPs show excellent optical performance with improved solar modulation property (*T*_sol_ = 34.3%) and luminous transmittance (*T*_lum, average_% = 53.9%).

## Figures and Tables

**Figure 1 nanomaterials-09-00970-f001:**
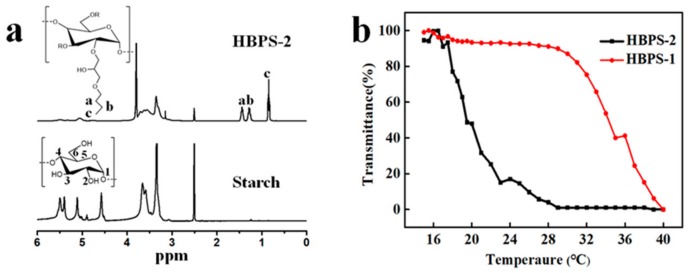
(**a**) ^1^H NMR spectra of soluble starch and HBPS-2; (**b**) Changes of transmittance at 550 nm vs. temperature for HBPS-1 (5 g/L, DS_BGE_ = 0.31) and HBPS-2 (5 g/L, DS_BGE_ = 0.42) recorded to determine the LCST value through heating process.

**Figure 2 nanomaterials-09-00970-f002:**
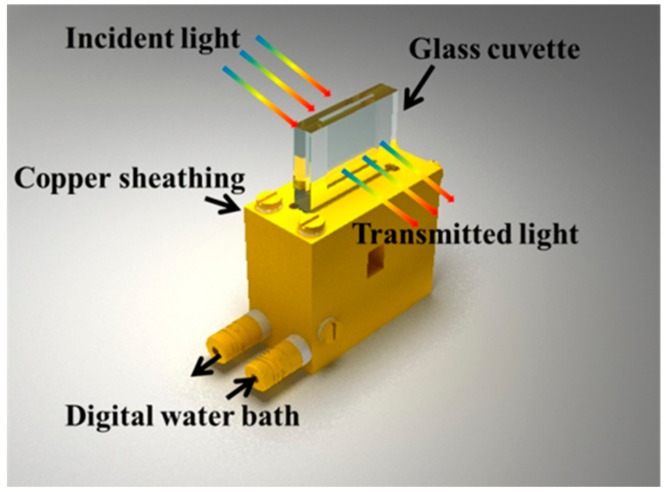
Schematics of the measuring device for the optical transmittance of samples

**Figure 3 nanomaterials-09-00970-f003:**
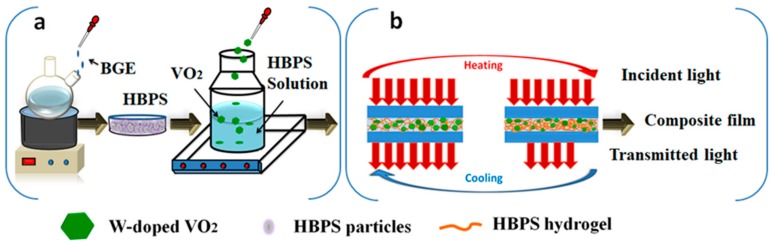
(**a**) The schematic illustration of the preparation of W-doped VO_2_/HBPS composite; (**b**) The structure and working principle of our hybrid thermochromic film. Solar light is mostly transmitted at lower temperatures (left) and partially blocked at higher temperatures (right).

**Figure 4 nanomaterials-09-00970-f004:**
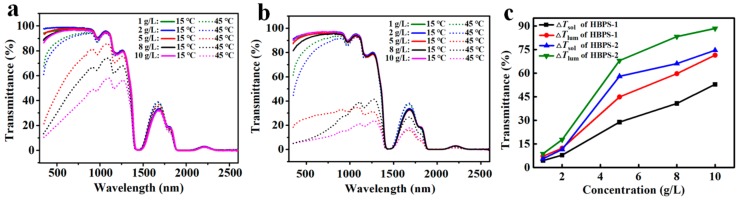
Transmittance curves of (**a**) HBPS-1 and (**b**) HBPS-2 with various concentrations between 1–10 g/L at 15–45 °C; (**c**) A summary for the optical performance of temperature-responsive starch derivative samples (HBPS-1 and HBPS-2) under different concentrations (HBPS-1 and HBPS-2 have different substitution degrees).

**Figure 5 nanomaterials-09-00970-f005:**
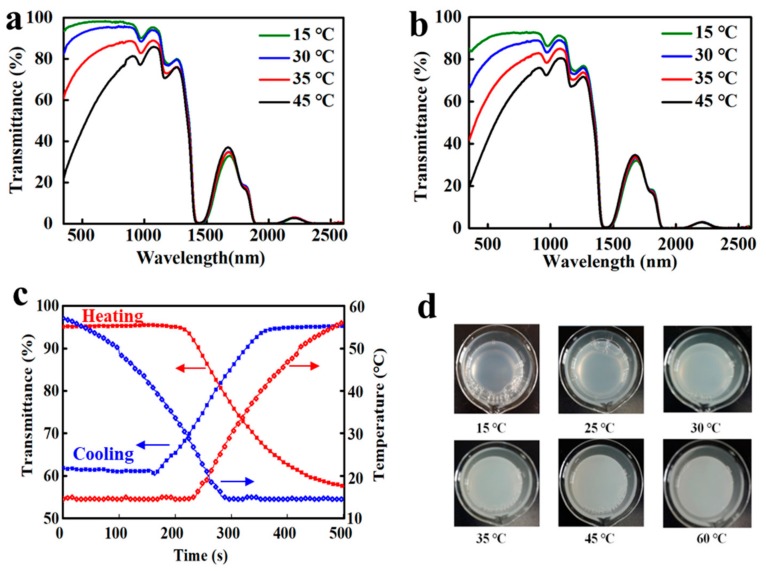
Optical transmittance spectra of (**a**) HBPS-1 (5 g/L) and (**b**) HBPS-1 (10 g/L) at different temperatures from 15 to 45 °C; (**c**) Temperature and transmittance (at 1100 nm) changes as a function of time through cooling or heating across LCST for HBPS-1 (10 g/L). The heating speed was 0.16 °C/s, and the cooling speed is 0.13 °C/s; (**d**) Pictures of HBPS-1 (10 g/L) at various temperatures.

**Figure 6 nanomaterials-09-00970-f006:**
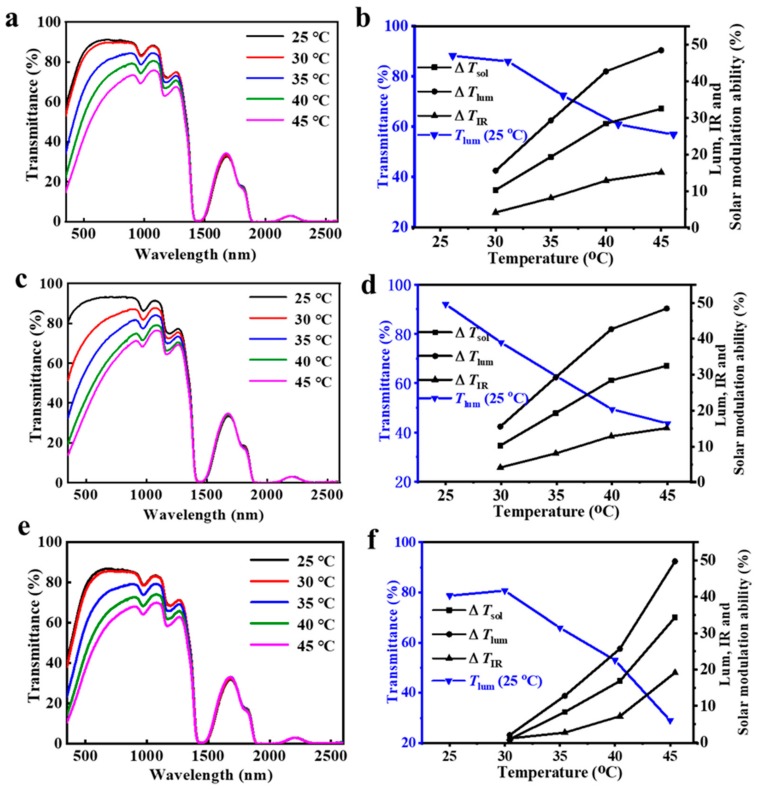
UV-vis near-IR spectra of (**a**) Composite 1, (**c**) Composite 2, (**e**) Composite 3 at different temperatures from 15 to 45 °C; changes of *T*_lum_, Δ*T*_sol_, Δ*T*_IR_, and Δ*T*_lum_ of (**b**) Composite 1, (**d**) Composite 2, and (**f**) Composite 3.

**Figure 7 nanomaterials-09-00970-f007:**
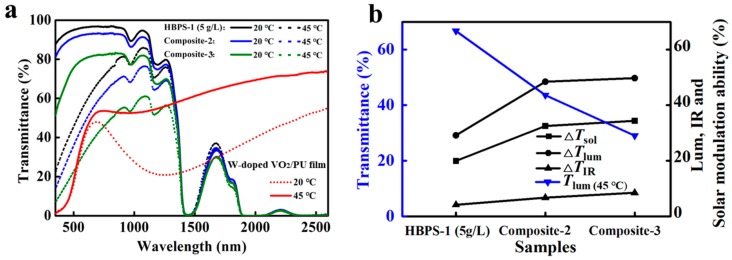
(**a**) UV-vis near-IR spectra of films of W-doped VO_2_ (M)/PU with 2‒3 μm thickness (red solid and dashed lines), HBPS-1 (5 g/L), Composite 1 and Composite 2 at 20 and 45 °C; (**b**) Changes of *T*_lum_, Δ*T*_sol_, Δ*T*_IR_, and Δ*T*_lum_ for films of HBPS-1 (5 g/L), Composite 2, and Composite 3.

**Figure 8 nanomaterials-09-00970-f008:**
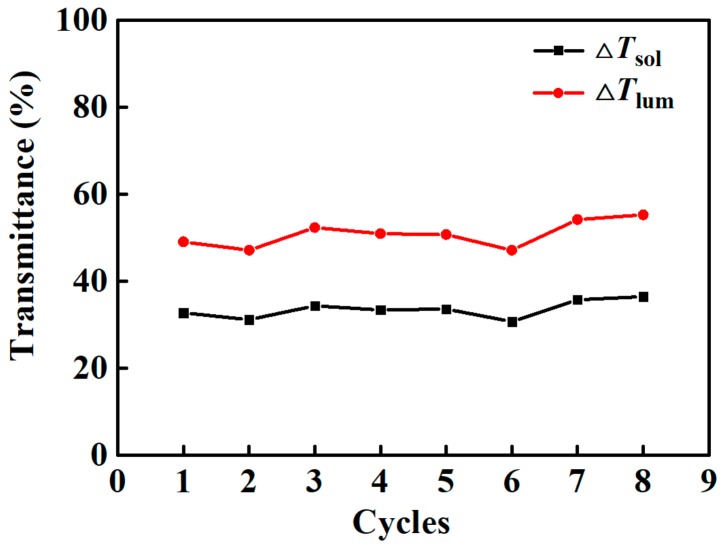
Changes in light transmittance at 1100 nm in the durability test of Composite 3 between 20–45 °C.

**Table 1 nanomaterials-09-00970-t001:** Description of the two different starch derivative hydrogels prepared in this work.

Sample	BGE:AGU ^1^	DS ^2^	LCST (°C) ^3^
HBPS-1	0.55	0.31	32
HBPS-2	0.65	0.42	21

^1^ Molar ratio (AGU means anhydroglucose unit); ^2^ DS, degree of substitution of butyl glucidyl ether (BGE) determined by ^1^H NMR; ^3^ Determined by the UV-vis spectroscopy measurement. HBPS: 2-hydroxy-3-butoxypropyl starch, LCST: low critical solution temperature.

**Table 2 nanomaterials-09-00970-t002:** Comparison of the optical properties between composites 1, 2, and 3 and poly (*N*-isopropylacrylamide) (PNIPAm) hydrogel, VO_2_/hydrogel hybrid, and VO_2_.

Sample	*T*_lum, low_ (%)	*T*_lum, high_ (%)	Δ*T*_lum_ (%)	*T*_lum, average_ (%)	Δ*T*_IR_ (%)	Δ*T*_sol_ (%)
PNIPAm hydrogel [11]	88.5	58.8	29.7	73.6	9.5	21.4
VO_2_/hydrogel [13]	82.1	43.2	38.9	62.6	29.9	34.7
VO_2_ [9]	45.6	40	5.6	42.8	---	22.3
HBPS-1 (5 g/L)	97.9	53.1	44.8	75.5	5.0	28.9
HBPS-1 (10 g/L)	95.8	49.6	46.2	72.7	6.3	31.0
HBPS-2 (5 g/L)	94.9	28.9	68.0	60.9	20.7	58.0
HBPS-2 (10 g/L)	96.2	7.8	88.4	52.0	26.2	74.6
Composite 1	88.2	56.9	37.6	72.6	11.6	24.7
Composite 2	92.0	43.6	48.5	67.8	15.1	32.5
Composite 3	78.7	29.0	49.8	53.9	8.5	34.3

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
