# Peer review of "Tungsten-Doped VO2/Starch Derivative Hybrid Nanothermochromic Hydrogel for Smart Window"

_nanomaterials, 2019, doi:10.3390/nano9070970_

Round 1

Reviewer 1 Report

The paper entitled "Tungsten-doped VO2/starch derivative hybrid nanothermochromic hydrogel for smart window" can be published in its current form.

Author Response

We are grateful to this reviewer for his/her positive comment on our manuscript.

Reviewer 2 Report

The numbering corresponds to my original comments. I still do not agree with the authors in all cases. I am leaving these cases for the editor to decide.

1) It is perfectly OK that the amount of VO2 nanoparticles in composites corresponds to 50-100 nm of pure VO2. However, the core of my comment was related to the thickness of pure VO2 which is also claimed to be 2-3 microns (red lines in Fig. 6a, please see the figure caption again), not to the composites.

2+3) OK, let's forget the sample HBPS-2, but the sample Composite-2 (which the numbers in Abstract and Conclusions are apparently based on) is not qualitatively different. Delta_Tlum is higher than Delta_Tsol also in this case. Again, I agree that the material may still be useful, but I do not agree that it makes sense to compare the performance with that of films with low Delta_Tlum where the transition takes place predominantly in the infrared.

The statement indicating that Delta_Tlum is not important because it can be decreased by lowering the thickness is relevant only if it can be decreased at preserved Delta_Tsol.

4) OK, improved

5) OK, improved

Author Response

Comment:

(1) It is perfectly OK that the amount of VO2 nanoparticles in composites corresponds to 50-100 nm of pure VO2. However, the core of my comment was related to the thickness of pure VO2 which is also claimed to be 2-3 microns (red lines in Fig. 6a, please see the figure caption again), not to the composites.

Response: We are sorry for using the incorrect description here. The red lines in Fig. 6a (now Fig. 7a in the new manuscript) are for the W-doped VO2(M)/PU film coated on PET with a total thickness of 2-3 μm. We have clarified this by using ‘W-doped VO2(M)/PU’ instead of ‘pure W-doped VO2’ in the figure caption and the main text.

Comment:

(2+3) OK, let's forget the sample HBPS-2, but the sample Composite-2 (which the numbers in Abstract and Conclusions are apparently based on) is not qualitatively different. ΔTlum is higher than ΔTsol also in this case. Again, I agree that the material may still be useful, but I do not agree that it makes sense to compare the performance with that of films with low ΔTlum where the transition takes place predominantly in the infrared.

Response: Thanks for this comment and we totally agree with you. To put it right, we have revised all the related statements throughout the manuscript. For example, we have deleted the sentences for such comparison in the Abstract, and also in the Conclusion of the revised manuscript. 

Reviewer 3 Report

I regret to inform the authors that despite they provided a revised manuscript with better quality than the original, the main issue is still present. I am referring to the number of experiments they carried out in order to determine the influence of W onthe VO2 composition and final properties. One concentration, determined only in alimentation (1%) is not enough to determine a general trend. I suggest to increase the number of experiments, i.e. by varying the amount of W attached on VO2 and provide effective evidences of their structural morphology, properties and dispersion behavior in the final composites.

Author Response

Thanks for your valuable comment. In this work, we investigated the properties of our composites mainly by adjusting the amount of VO2 NPs (W-doped) in the final composite material, NOT the doping amount of W in the VO2 NPs that will then disperse in the composite material.

First, the major factors affecting the optical performance of our composites are the concentrations of starch and VO2 (W-doped) NPs, while the doping amount of W in VO2 NPs mainly influences the phase transition temperature of NPs and thus the composites. As for the dispersion of W-doped VO2 NPs in composite material, we applied the grinding method (Figure S2 in SI) to the VO2 NPs prior to mixing them with starch to avoid the precipitation of VO2 NPs and achieve good dispersion. Therefore, in our opinion, the content of VO2 NPs, rather than the doping content of W in the NPs, has a much more significant effect on the morphology, optical performance and NP dispersion behaviour in our final composites.

Second, we have previously studied and reported the optimum recipe for the preparation of W-doped VO2 NPs, including the optimum W doping amount, to achieve the best optical performance of VO2 NPs (Chen et al. J. Mater. Chem. A. 2014, 2, 2718-2727; Shen et al. Phys. Chem. Chem. Phys., 2016, 18, 28010-28017). Based on these results, in this work, we employed the same recipe to prepare the W-doped VO2 NPs with the purpose of obtaining the optimized W doping amount (~1%) we found previously. And DSC result (Figure S1 in SI) confirmed that we successfully prepared W-doped VO2 NPs with a phase transition temperature of 40 oC and a W doping content of ~1 at%. Actually it was our target to obtain such VO2 NPs undergoing phase transition at near-room temperature for the subsequent preparation of our composite material. This means changing the W doping content in VO2 NPs will miss the mark.

Third, we did the relevant experience and discussed the effect on the usage of the VO2 content to the optical performance of the composite after we selected a proper usage and content of the starch. We re-added the detailed description of the effect of the W-doped usage on the optical performance of the composite.

In sum, we think it is a more direct and effective approach for the optimization of the solar modulation property of our final composites by adjusting the content of the VO2 NPs (W-doped) in the composite, rather than the W doping content. To illustrate this point, we have added another composite sample in Table 2 of the revised manuscript and compared three composite samples (Composites-1, -2 and -3) with different W-doped VO2 NPs contents in Section 4.3.

Round 2

Reviewer 2 Report

I recommend the corrected manuscript for publication.

This manuscript is a resubmission of an earlier submission. The following is a list of the peer review reports and author responses from that submission.

Round 1

Reviewer 1 Report

I believe that the paper entitled "Tungsten-doped VO2/starch derivative hybrid nanothermochromic hydrogel for smart window" can be published in Nanomaterials after taking into consideration the following minor points:

You need to include the following references along with [4-8]

Solar Energy Materials and Solar Cells 140 (2015) 1-8
Physics Procedia 46 (2013) 137-141

I suggest not to start the sentence with numbers in "2.2. Preparation of HBPS and HBPS Hydrogel Films".

Please remove "were obtained" in the sentence below

"The LCST value of HBPS hydrogel could be controlled by varying the dosage of BGE [17],
HBPS-1 and HBPS-2 (Table 1) with different LCST values were obtained."

It is "several" and not "serval".

Please remove "were successfully made" in the sentence below

"Composite-1 and Composite-2 synthesized with different addition amounts of
W-VO2 were successfully made (see Supporting Information Table S1)."

Please remove "was recorded" in the sentence below

"The response behavior of the prepared samples was monitored also by the UV-vis-near-IR
spectrometer with a water bath to control the system temperature and transmittance at 1100 nm as a function of time during the switching process was recorded."

Please rewrite the following sentence. English grammar is not correct.

"The more hydrophobic groups there are being grafted on the starch backbone, the
lower LSCT value HBPS has [21]."

It is "thermo-responsive" not "thermos-responsive".

Reviewer 2 Report

The manuscript deals with several composite films containing thermochromic W-doped VO2 nanoparticles. My overall impression is that manuscript is far from acceptable. Although the results may have some merit, presenting them correctly would require so many changes that I would call them writing a new paper from scratch rather than a revision.

(1)

Regarding the information that the films are 2-3 microns thick: I am not sure about the composites, but regarding the pure VO2 (red lines in Fig. 6a) this information is outside reality. Approximate optical constants of VO2 are well known, and the results in Fig. 6a clearly correspond to a thickness on the order of tens of nanometers, not microns.

(2)

The whole point of the worldwide efforts in the field of VO2, as I see it, is the transmittance modulation in the infrared (without affecting the visible transmittance, i.e. high delta_Tsol at low delta_Tlum). This is opposite to what the authors did: Figure 4 clearly shows that what the authors are really studying is transmittance modulation in the visible (delta_Tsol is actually systematically lower than delta_Tlum!). I do not exclude that such materials (acting as good old sunblinds and often called photochromic rather than thermochromic) may still be interesting, but it makes zero sense to compare their properties with the properties of materials which preserve Tlum (see also the next comment). 

(3)

I agree that using a single value of average Tlum (below and above the transition temperature) constitutes a useful criterion of success when describing films which exhibit low delta_Tlum. However, when describing the properties of author's films (changes of Tlum from 95% to 29%, from 96% to 8%, etc.), this is just confusing. I guess that what matters for the people inside a building with smart windows is the minimum value of 8% rather than the average value of  (96+8)/2 = 52%.

(4)

A potentially strong point of the manuscript is decreasing the transition temperature by doping VO2 by W. However, there are numerous related weaknesses

- the doping is not sufficiently reflected in the Introduction: I suggest to provide transition temperatures achieved in the cited papers, and add representative citations of papers using the same doping previously (recent examples doi.org/10.1007/s10971-015-3832-z, doi.org/10.1063/1.4979700, doi.org/10.1016/j.solmat.2018.12.004)

- the W content is not known ("a certain amount of W dopants were added"), which makes the results difficult to reproduce

- the width of hysteresis curves is not known (the heating branch and the cooling branch are typically different, and I do not even know which one is shown in Fig. 3b)

(5)

The fact that the authors checked the performance stability is another potentially strong point, but I am not convinced that Fig. 7 shows "high performance stability". High stability according to which criterion, compared to what? Especially the changes of delta_Tlum look actually quite large to me.

Reviewer 3 Report

In the present manuscript, the authors reported the preparation of W-doped VO2 nanoparticles and their incorporation within starch derivatives for the preparation of hybrid hydrogels with thermochromic characteristics. The paper reports new results with respect to a similarly published manuscript in 2018 by the same authors: "Thermo- and pH-responsive starch derivatives for smart window" Carbohydr. Polym. 2018, 196, 209-216. Notwithstanding the introduction of W-doped VO2 nanoparticles that potentially offered the main novelty of the manuscript I cannot find any relevant evidence for considering the manuscript suitable for the journal. Some parts including the experimental part, were reported without providing useful information to reply the experiments. The nature of the nanoparticles, the level of doping, their average size and morphology before and after the introduction in the matrix were missing. Since it represents the main goal of the s manuscript, I suggest the rejection of the paper and its resubmission after strong revision. Moreover:

1) at pag 3 line 120 and pag 4 line 138, the authors used the term solution instead of dispersion. The nanoparticles are supposed to be insoluble in the medium;

2) the coating procedure as well as the typology of the coating machine should be added;

3) Please explain the mechanism behind the formation of HBPS hydrogels. What is the cross linking degree?

4) please report the concentration in the caption of figure 3b;

5) Figure 4 shows the optical properties of the films but it's not clear what "concentration" stands for. 

Reviewer 4 Report

The article “Tungsten-doped VO 2 /starch derivative hybrid nanothermochromic hydrogel for smart window” by Wang et al. reports about hybrid films prepared from HBPS (thermo-responsive material with low critical solution temperature) with dispersed W-doped VO2 NPs with highest solar modulation property and average luminous transmittance.  The authors explain in detail the hybrid composite preparation method. They prepare HBPS samples with different thermo-response properties (2 different starch derivative hydrogels) and use one of them to prepare the hybrid organic-inorganic composite films.  UV-vis-near-IR measurements are performed on the HBPS samples and the composites.  A durability test of the transmittance of one of the composites at 1100 nm is provided. The authors conclusions are directly derived from the reported data.